# Exploring the Potential of Alternative Materials in Concrete Mixtures: Effect of Copper Slag on Mechanical Properties and Carbonation Resistance

**DOI:** 10.3390/ma16206677

**Published:** 2023-10-13

**Authors:** Yimmy Fernando Silva, Mónica Villaquirán-Caicedo, Silvia Izquierdo

**Affiliations:** 1Concrete Innovation Hub UC (CIHUC), Composite Materials Group (CENM), School of Civil Construction, Faculty of Engineering, Pontificia Universidad Católica de Chile, Santiago 7820436, Chile; 2Composite Materials Group (CENM), School of Materials Engineering, Universidad del Valle, Cali 760042, Colombia; monica.villaquiran@correounivalle.edu.co; 3Development Engineer, Cemex, Bogota 13001, Colombia; silviaraquel.izquierdo@cemex.com

**Keywords:** copper slag, concrete, carbonation, compressive strength, residues

## Abstract

In this study, the effect on the flowability, compressive strength, absorption, sorptivity, and carbonation resistance of concrete with different copper slag (CS) replacement ratios was investigated. For this research, four concrete mixes with different percentages of CS were made (0%, 10%, 20%, and 30% of CS as replacement of cement by volume). In addition, the microstructure was analyzed by X-ray diffraction (XRD), scanning electron microscope (SEM), and thermogravimetric analysis (TG–DTG). The results shows that the incorporation of CS reduces the workability and compressive strength of the mixtures, being more significant in concrete with 30% CS. The carbonation depth of concrete with CS increases monotonically with increasing CS. In addition, the compressive strength of the carbonated (20% and 30% CS) concretes show a loss of compressive strength at 90 days of exposure when compared to their water-cured counterparts. The use of low percentages of CS does not generate a decrease in workability and its mechanical effect is not significant at prolonged ages, so the use of this waste as SCM in percentages close to 10% is a viable alternative to the sustainability of concrete and the management of this residue.

## 1. Introduction

The construction industry is one of the sectors that generates the highest carbon dioxide emissions and consumes natural resources [1,2,3,4]. Additionally, it contributes to generating approximately 50% of global waste streams [5]. Therefore, ensuring the sustainability of the construction industry is crucial, which can be achieved through reduced consumption of natural resources and a smaller carbon footprint. Consequently, the use of supplementary cementitious materials (SCM) has become an efficient method to reduce CO_2_ emissions [6,7], primarily because it reduces the usage of cement.

The use of SCM in concrete can improve the performance of this composite material in terms of mechanical properties, workability, and durability [8]. This improvement in concrete performance is attributed to the pozzolanic behavior of SCM, which makes it suitable for different concrete mix designs. SCMs can be natural or by-products of industrial processes (mining or agricultural). The ASTM C618 [9] specifies certain requirements for natural pozzolans or fly ashes, among which are that the amount of SiO_2_, Al_2_O_3_, and Fe_2_O_3_ must be at least 50% and 70%, respectively. However, this standard guide limits the classification of SCMs to two materials (natural pozzolans and fly ashes), so an alternative for evaluating new SCMs is the process described in ASTM C1709 [10]. 

In general, SCMs with pozzolanic activity are used as a partial substitute for cement to generate pastes. In this binder, SCMs generate the pozzolanic effect and the dilution effect, which leads to a reduction in calcium hydroxide (Ca(OH)_2_) [11,12], which is produced in the cement hydration process (Equation (1)). The reduction in Ca(OH)_2_ is due to SCM reacting with Ca(OH)_2_ and forming secondary C–S–H (Equation (2)) [11,13].
Cement + H_2_O→C–S–H gel + Ca(OH)_2_(1)
Ca(OH)_2_ + SCM→C–S–H secondary gel(2)

### 1.1. Literature Review

Copper slag (CS) is a solid waste with high hardness produced from the production and refining process of copper (an indispensable metal in the modern industry [14]) and is mainly composed of F_2_O_3_ and SiO_2_ in percentages between 35–60% and 25–40%, respectively, in addition to few amounts of Al_2_O_3_, CaO, and CuO [15]. In copper production, for one ton of crude copper, between 2.2 and 3.0 tons of copper slag are produced, reaching 40 million tons in 2020 worldwide [16]. The high generation of this by-product generates major environmental impacts such as water and soil contamination [17], and if it is accumulated for a long time without treatment, it destroys land resources [18].

To reduce the environmental impact generated by CS, several studies have been carried out to take advantage of this by-product. CS has been used as an aggregate in the production of asphalt mixtures for several years. For example, Rojas-Pardo et al. [19] incorporated CS in different sizes (2.00 mm, 0.25 mm, <0.063 mm) as a partial replacement of natural aggregate to evaluate its effectiveness as a conductive material, obtaining self-healing indexes in mechanical resistance higher than 40% in all sizes, presenting the highest index (60%) in the mixture that used a size of 2.00 mm. Another study was conducted by Raposeiras et al. [20], who evaluated the tensile strength of asphalt with different copper slag sizes (between 2.5 and 0.08). Different authors have observed that the strength increased as the particle size increased, being higher than the reference mix when the 2.5 mm size was substituted. Additionally, slag has been studied in Portland cement matrix mixes as fine and coarse aggregates, showing improvements in mechanical performance [15,21]. However, when CS has been used as coarse aggregate, there are contradictory results [22,23]. Choudhary [22] used replacement volumes between 20% and 100% of the natural coarse aggregate with CS, observing a decrease in compressive and tensile strength of 39% and 37.5%, respectively. However, when Lori [23] used CS as coarse aggregate, compressive strength increased by 22% when 100% of the natural coarse aggregate was substituted.

Research on CS as a supplementary cementitious material (SCM) has garnered attention recently due to the importance of decarbonizing the cement industry [24]. For example, Vizcaino and Silva [25] used CS as an SCM in a preliminary study on mortars, observing a decrease in compressive strength at all percentages used (10–50% by volume) at early ages (7 and 28 days of curing). However, at 150 days of curing, the mortar with 10% CS showed an 8.1% increase compared to the reference mixture. He et al. [26] reported the application of CS as a partial replacement (10–50 wt%) by Portland cement in pastes. They also noted a loss of compressive strength of 85% and 33% at 3 and 28 days, respectively, in the paste with 50% CS compared to the composite without CS. 

### 1.2. Significance and Content of This Study

The present research seeks to fill the gaps that exist in the use of CS as a SCM, by evaluating the mechanical performance and carbonation resistance of concrete. This last property is becoming more and more relevant due to the fact that the presence of CO_2_ in the environment has increased in the last years, reaching a monthly mean CO_2_ concentration of 416 ppm in March 2021 [27]. The CO_2_ from the atmosphere is considered to be one of the aggressive substances for reinforced concrete, since it can diffuse through the pore network of concrete, dissolve in the pores and react with the portlandite or C–S–H. This phenomenon is known as carbonation [28]. This study presents the influence on the properties in both fresh and hardened states of concretes with different levels of replacement (0–30% by volume) of CS by Portland cement in concrete. The properties in fresh (slump) and hardened state (compressive strength, density, absorption, porosity, carbonation resistance) were evaluated. Simultaneously, a microstructural analysis was conducted using thermogravimetric analysis (TGA–DTG), scanning electronic microscopy (SEM), and X-ray diffraction patterns (XRD). The results of this study will contribute to the generation of knowledge about the use of CS in concrete and its performance in environments with high concentrations of CO_2_, which are becoming increasingly common in our environment.

## 2. Materials and Methods

Hydraulic cement of general use of the company Argos, type GU according to ASTM C1157 [29], was used. Its chemical composition was determined by X-ray fluorescence (XRF) and is reported in Table 1, and the morphology is shown in Figure 1, where particles of different sizes and irregular shapes are observed. The CS was obtained from a company dedicated to the exploitation of copper in Chile, which has a density of 3.55 g/cm^3^. This by-product was crushed and milled to be used as a supplementary cementitious material; the morphology and particle size distribution of the CS and cement are shown in Figure 2 and Figure 3, respectively. The particular range of this by-product of the mining industry was 0.55–120 µm, the average size being 24.94 µm. The mineralogical phases of CS were identified by X-ray diffraction (scanning range: 5–60° (2θ)) and are shown in Figure 2, where it can be identified that the predominant crystalline phases are fayalite and magnetite, which is consistent with the chemical characterization obtained by XRF (Fe_2_O_3_: 55.58% and SiO_2_ 20.43%). 

Natural river sand was used as fine aggregate and crushed gravel with a nominal maximum size of 12.7 mm was used as coarse aggregate. The physical characterization of aggregates is given in Table 2. A polycarboxylate-based superplasticizer (SP) was also used to improve the workability of the concrete without using high water/cement (w/c) ratio.

### 2.1. Mixtures

The concrete mixtures were designed according to ACI 211 guidelines. Four levels of replacement of CS by cement were made (including the reference concrete). Table 3 shows the mixture proportion of all the concretes carried out in the present investigation. The content of water, sand, gravel, and SP remained constant in all the mixtures, and the cement was replaced in 0%, 10%, 20%, and 30% in volume of CS.

The concrete specimens were cast using a revolving drum mixer. The mixing process remained constant in all the concrete mixes carried out. Before using the mixer, a manual homogenization of the fine material (cement–CS) was carried out for concretes that contained CS in their composition. First, the aggregates (coarse and fine) were mixed with approximately a quarter of the total water for two minutes. Then, the binder (cement only (reference mix) or cement with CS) was added with half the water and mixed for two minutes, without interrupting the mixing process, before the remaining water mixed with superplasticizer was added and mixed for two more minutes.

### 2.2. Test Methods

Four tests were performed on the concrete to study the influence of slag content in the mixtures. The slump flow test [30] was carried out to investigate the influence of CS on the workability of concrete mixes added with this by-product. The test was performed by filling the cone in two layers and tamping each layer 25 times. The compressive strength of concretes were tested on hydraulic press (Controls-CvTech 1500) according to ASTM C39 [31]. Three samples were prepared (7, 28, 56, 84, and 118 days of curing in water) for each age evaluated and mixing performed. The compressive strength of the concrete was also evaluated in the accelerated carbonation chamber after 28, 56, and 90 days of exposure. The density, void content, and water absorption of concrete after 28 days of curing were measured according to ASTM C 642 [32]. For the resistance to carbonation test, the concrete mixtures and cylindrical specimens of 76.2 × 15.24 mm were used, which were cured for 28 days under water. Then, the top and bottom of the cylinders were sealed and placed in a carbonation chamber under controlled conditions (1.2% of CO_2_, 23 ± 2 °C, 65 ± 2% RH). At 28, 42, 56, 70, and 90 days of exposure, each cylinder was cut and the depth of carbonation was measured by applying phenolphthalein to the surface and the depth of carbonation was measured (six measurements were taken per specimen). The cylinders were returned to the carbonation chamber for continued accelerated CO_2_ exposure after resealing the top of the remaining cylinder.

## 3. Result and Discussion

### 3.1. Fresh State (Slump)

Results of the slump test of the four concrete mixtures are shown in Figure 4; all the fresh concrete show good consistency and homogeneity. It can be observed in Figure 4 that at high amounts of substitution of CS for cement in the concretes, a loss of workability is generated, being greater in the 30% CS mix. The 20% SC and 30% SC mixes show a decrease in the slump of 54% and 73%, respectively. However, the 10% SC shows a higher slump value than the reference mix. The improvement in the workability of the mix with 10% CS could be attributed to the low water absorption of this by-product. In addition, the CS could fill the spaces between some cement particles, which will produce a reduction in the interstitial water, which would increase the fluidity [33,34]. However, for the slump loss of the concretes with the higher percentages of CS in their design, this may be due to their filling effect reaching its limit, in addition to the irregular particle morphology that significantly influences the yield stress (increased cohesion and yield stress) [33]. A similar behavior was found by Ahad Barzegar (2022) [35], who found that the loss of workability in concrete with CS incorporation, due to its morphology, makes a greater amount reached of superplasticizer in the mixes, to achieve similar consistency between concretes.

### 3.2. Compressive Strength

The compressive strengths of all the mixtures at different ages of curing are shown in Figure 5. It is shown that replacement of cement by CS in concretes generates a monotonic decrease in compressive strength as the CS content increases; this is related to the dilution effect due to the lower cement content of the CS mixtures. The higher average strength of the 10% CS concrete at 7 days of curing can be attributed to the filling effect of the CS particles, generating a reinforcement in the densification of the interface zone in this mix [36]. Additionally, the CS particles had angular morphologies, which can improve the cohesion in the concrete matrix [37], which was evident in the strength developed at this age of curing. At the curing period of 28 and 118 days, the reference mixture exhibited the highest compressive strength; and all the mixes were ranging from 28.12 to 17.17 MPa, and from 30.1 to 23.65 MPa. At 28 days, the difference between the reference concrete and the concrete with 30% CS was 38.9% and 21.4%, respectively, which means it could be said that the CS generates a positive effect on the mechanical strength at prolonged curing ages. Similar results have been found by Ahad Barzegar (2022) [35], where the presence of large amounts of mine tailings and heavy metals, cement hydration process rate, and C–S–H crystals growth were low.

### 3.3. Water Absorption, Apparent Density, and Voids

The water absorption, apparent density, and void result for 28 days of aging of the reference and concrete with CS are presented in Table 4. As shown, the apparent density at the two curing ages evaluated increased monotonically when the slag content was higher in the mixtures. The water absorption and porosity of hydrated concrete at 28 days increase in mixtures with high percentages of slag (20 and 30% of CS), and this increase is attributed to the configuration of the pore system [38]. On the other hand, the lower absorption and porosity of the 10% CS mix are attributed to the filling effect generated by a better accommodation of the concrete with this percentage of CS in the mixture.

### 3.4. Susceptibility to Accelerated Carbonation

All concretes exposed to accelerated carbonation conditions were previously cured for 28 days in water. After this period, specimens of each concrete were placed in the carbonation chamber with a concentration of 1.2% CO_2_, temperature of 23 ± 2 °C, and relative humidity of 65%. The depth of carbonation was determined using phenolphthalein, as shown in the upper images of Figure 6. These images correspond to the different concretes exposed in the carbonation chamber for 28 days. The carbonated zone after the application of phenolphthalein shows no change in coloration, while the non-carbonated zone turns purple (the area farthest from the exposure surface). In this figure, an analysis was also performed using the Matlab 9.5 software to determine the non-carbonated area of the evaluated surface of the reference concrete, 10% CS, 20% CS, and 30% CS, being 31.3%, 22.78%, 19.15%, and 15.26%, respectively. Additionally, Figure 7 shows the carbonation depth of all concretes at different exposure periods. It is evident that the reference mixture, which has the highest cement content in its design, shows a higher resistance to carbonation, as opposed to the 30% CS concrete, which has the lowest resistance to carbonation, where after 90 days of exposure in the carbonation chamber it is already totally carbonated. The higher resistance of the reference mix is attributed to its higher content of clinker, which results in higher content of portlandite, resulting in a higher alkaline reserve or greater carbonatable material, and, therefore, higher resistance to carbonation [39]. In concretes with high CS content, the carbonation resistance decreases due to the higher porosity, and the lower amount of cement hydration products (dilution effect) in these mixtures generates a lower alkaline reserve [26,40]. This behavior is similar to that observed by Silva and Delvasto [40], who used residue of masonry (RM) as SCM in self-compacting concrete and subjected it to accelerated carbonation. It was evident that with higher percentages of cement replacement, carbonation increased, attributed to the lower alkaline reserve of the concrete, as well as the low reactivity of the RM.

#### Effect of Carbonation on Compressive Strength

Figure 8 shows the compressive strength of the different mixes cured for 28 days in water (initial) and exposed to CO_2_ for 28, 56, and 90 days. At 28 days of exposure, there is a gain in compressive strength of 37.8%, 27.2%, 52.7%, and 48.3% of the reference, 10% CS, 20% CS, and 30% CS mixtures, respectively, compared to the compressive strength developed of the concretes before to the beginning of the exposure in the carbonation chamber. This behavior is attributed to the carbonation reaction of CO_2_ with Ca-rich minerals, especially portlandite [41], as this generates a denser microstructure, which leads to greater strength, as well as making it more difficult for CO_2_ to penetrate the samples. However, at 90 days of exposure in the carbonation chamber of the different mixes, the compressive strength decreased when compared to that developed at 28 days of exposure to CO_2_. Despite this loss in compressive strength of the different concretes, the strength at 90 days of exposure of the different concretes is not lower than the strength of the mixtures at 28 days cured in water (initial). The loss in compressive strength of the different mixtures at long periods of exposure to CO_2_ is attributed to the decomposition of calcium silicate hydrate (C–S–H) and the formation of micro-cracks [42]. 

Figure 9 shows the relationship between the mixes cured in water and exposed to accelerated carbonation for the same time. The mixes during the first 28 days of exposure in the carbonation chamber compared to the samples for 28 days cured in water (56 days total of water curing) show higher compressive strength. However, the concretes with the highest SC contents at 90 days of exposure in the carbonation chamber compared to their counterparts at 90 days cured in water (118 days total cured in water) show lower compressive strength, being 7.02% and 14.5% for 20% CS and 30%, respectively. The reduction in compressive strength was explained previously.

### 3.5. Microstructural Analysis

#### 3.5.1. Thermogravimetric Analysis

Figure 10 shows the analysis performed on samples of cement pastes curing in water and after of 28 days exposure to accelerate carbonation. Xian and Shao (2021) [43] and Rostami et al. (2012) [44] report that weight loss of the sample referred to the decomposition of C–S–H, ettringite, and gypsum between 105 and 200 °C, and related to the decomposition of C_3_A and C_4_AF hydration products. From 200 °C to 450 °C corresponds to the dehydration of calcium hydroxide Ca(OH)_2_ [45]. In total, the mass loss between 105 °C and 540 °C signified the weight loss of combined water of all hydration products subject to thermal decomposition (Table 5). As the temperature increased higher from 540 to 720 °C, poorly crystallized calcium carbonates (including amorphous calcium carbonates) would be decomposed. leading to the mass loss of the sample, while the decarbonization of better crystallized calcium carbonates took place above 700 °C [44,45]. Comparing the total mass loss in the non-carbonated sample, which was higher than that of carbonated reference sample, it indicates that a large amount of calcium hydroxide existed in the reference sample before carbonation treatment [46]. After 28 days accelerated carbonation, the TGA curves of REF and 20% CS paste show a rapid mass loss below 200 °C, indicating the dehydration of C–S–H. There is a clearer sudden drop of the TGA curve corresponding to an obvious peak of the DTG curve at around 425 °C in the reference paste, and 418 °C in 20% CS paste, suggested that calcium hydroxide was consumed during test carbonation. This is in accordance with XRD results (Figure 11). However, the reference carbonated sample (Figure 10A) mass kept losing after 450 °C to 500 °C, implying that some hydration products or some calcium hydroxides in their transition status to calcium carbonates were continuing to be decomposed during this elevated temperature [43]. 

#### 3.5.2. X-ray Diffraction (XRD)

The XRD patterns of the reference mix (100% cement) and 20% CS at 28 days of curing and 28 days after exposure in the carbonation chamber are shown in Figure 11. These diffractograms help to understand the main effects that CO_2_ has when in contact with cementitious matrix materials. Figure 11a shows the hydration products with crystalline structures such as ettringite and portlandite (CH) [47,48], produced in the dissolution and reaction of C_3_A with gypsum and the hydration of C_3_S and C_2_S, respectively, as well as the non-hydrated phase C_2_S, and quartz. After 28 days of accelerated exposure in the carbonation chamber, the CH peak becomes less intense due to the carbonation process and generates a higher amount of CaCO_3_. It is important to note that no diffraction peaks of the C–S–H gels were identified due to their glassy nature. Figure 11b shows the XRD patterns of the 20% CS mixture cured in water and exposed in an accelerated carbonation environment for 28 days, where the same phases of the reference mixture were identified, in addition to the characteristic phase of the EC such as fayalite. It can also be observed that the phase such as CH presents a lower intensity at 28 days of curing compared to the reference mix due to the dilution effect, since the 20% CS mixture presents a lower amount of C_2_S and C_3_S that generates C–S–H and portlandite. With respect to the carbonation process, it is evident that the 20% CS sample in the carbonation chamber (CO_2_) generates a higher amount of CaCO_3_, which can be generated from the CH and Ca of the C–S–H [49].

#### 3.5.3. Scanning Electron Microscopy (SEM)

In Figure 12 can be observed the pastes curing in water during 28 days, and after 28 days in an accelerated carbonation test. For the water-cured pastes, the presence of microcracks can be distinguished, in which the formation of crystalline products associated with ettringite waters can be observed, which are larger for the 20% CS paste. The smoother and more homogeneous surface is associated with the reference paste. When comparing the pastes after 28 days in the accelerated carbonation chamber, it can be observed that the most homogeneous surface is still in the reference paste, associated with the presence of hydrated calcium silicate gels (C–S–H). Greater heterogeneity and porosities were found on the surface of the 20% CS paste. The C–S–H is the main compound formed during hydration as explained in the previous section, and it is characterized by a calcium to silicon ratio (Ca/Si) between 0.6–1.6 [50] and structurally associated to the tobermorite gel. Using EDS analysis (Table 6), it was found that, in general, all Ca/Si molar ratios are greater than 1.6, indicating, according to [50], the presence of calcite and vaterite. The yellow and red numbers on the SEM micrographs correspond to the areas where EDS is performed. Likewise, the presence of high carbon content was found for both environments, however, it was higher for the samples exposed to accelerated carbonation, corroborating XRD results. Ca/Si ratios highly influence the mechanical performance of concretes. For tests performed on cement pastes, it has been found that as the Ca/Si ratio increases, the volume of the C–S–H gel decreases and, therefore, loss in compressive strength is expected [45].

## 4. Conclusions

In this study, the effect of the use of copper slag (CS) on the slump, compressive strength, porosity, sorptivity, and carbonation resistance, and its effect on microstructure, were investigated. Based on the results, the following conclusions were drawn:

The use of CS as a replacement for cement in concrete mixtures has been investigated, and it has been found that the workability of the concrete decreases as the percentage of CS increases. The irregular particle morphology of CS can affect the cohesion of the concrete, and the use of superplasticizers may be necessary to achieve similar consistency between concretes.

The replacement of high percentages of CS (20% and 30%) by hydraulic cement generated a loss in the slump compared to the reference concrete (100% cement).

The increase in CS in the concrete designs monotonically reduces the compressive strength, with the mix with 30% CS having the lowest strength. This behavior is attributed to the dilution effect generated by substituting CS for cement

The concrete mixture with CS presented a greater susceptibility to carbonation, due to the lower alkaline reserve (presence of portlandite) attributed to the dilution effect, indicating that the CS used does not provide a good performance in the concrete mixtures in this environment. The use of CS generated a higher carbonation rate, which increased monotonically as a higher CS content was used. However, the incorporation of CS can improve the compressive strength of the concrete specially at prolonged curing ages, due to the denser microstructure generated by the carbonation reaction of CO_2_ with Ca-rich hydration products.

The incorporation of CS in low percentages as a SCM (around 10%) improves flowability, enabling the production of concrete mixtures with a lower water–cement ratio while achieving the same consistency as a concrete mixture with 100% cement. This could compensate for the slight reduction in compressive strength at early ages, which is not very significant at longer curing ages.

## Figures and Tables

**Figure 1 materials-16-06677-f001:**
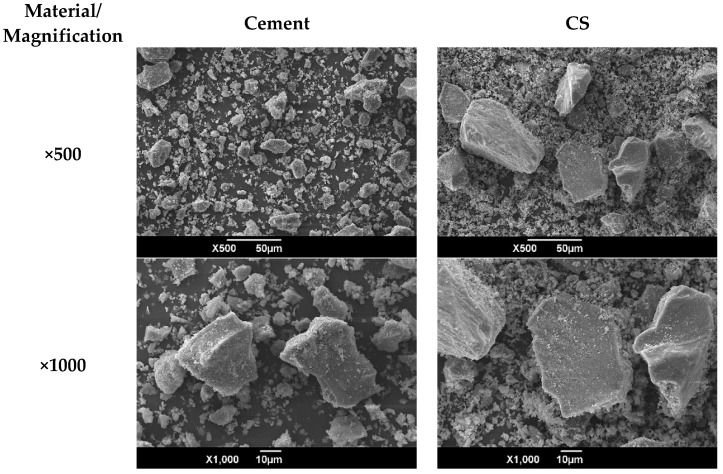
Morphology of cement particles and CS.

**Figure 2 materials-16-06677-f002:**
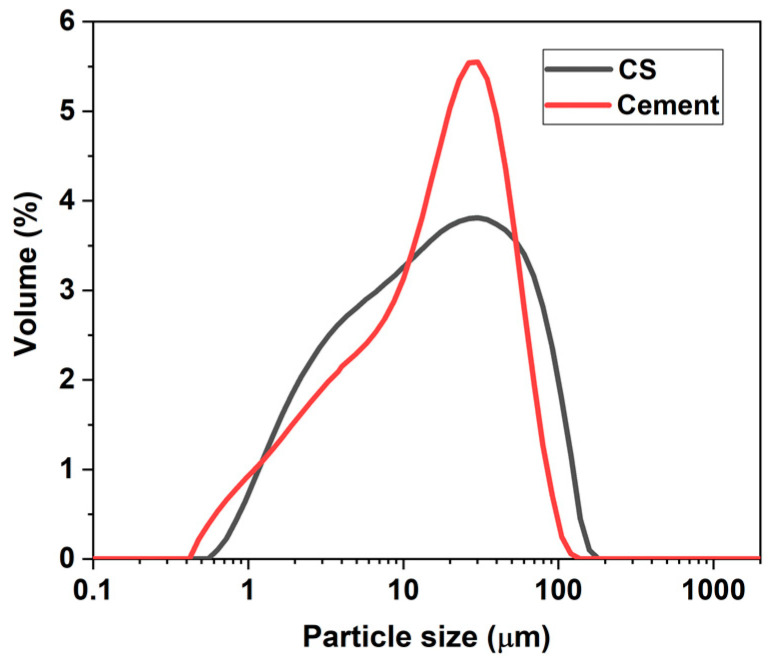
Particle size distributions of cements and CS.

**Figure 3 materials-16-06677-f003:**
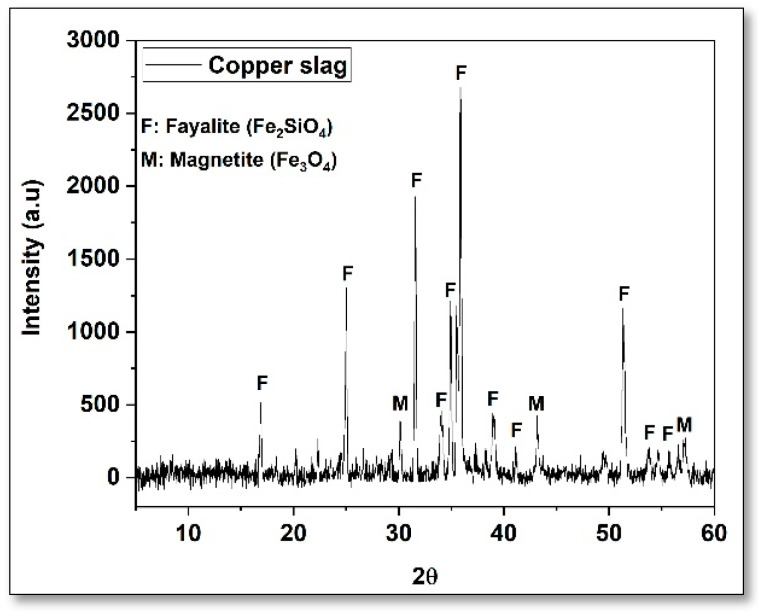
Mineralogical composition of CS (XRD).

**Figure 4 materials-16-06677-f004:**
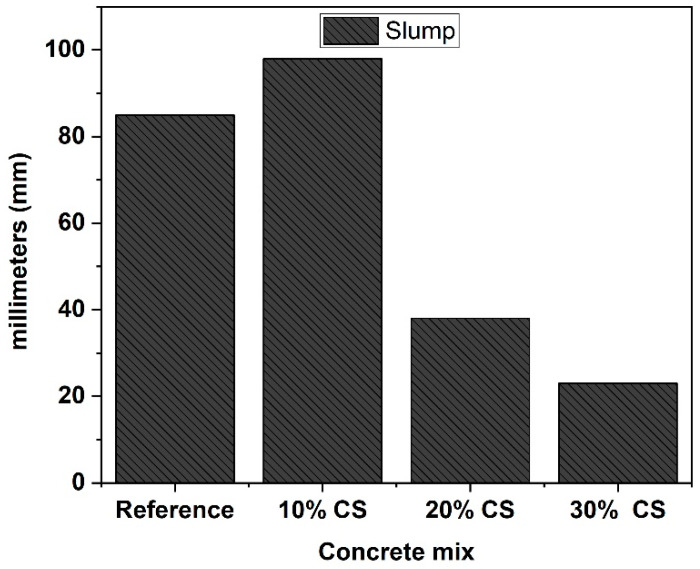
Slump of concretes with different concentrations of CS.

**Figure 5 materials-16-06677-f005:**
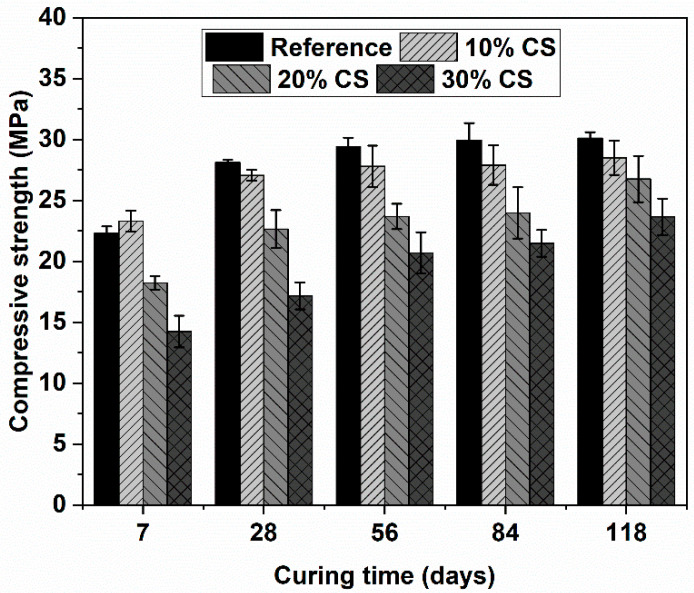
Compressive strength of water-curing concrete.

**Figure 6 materials-16-06677-f006:**
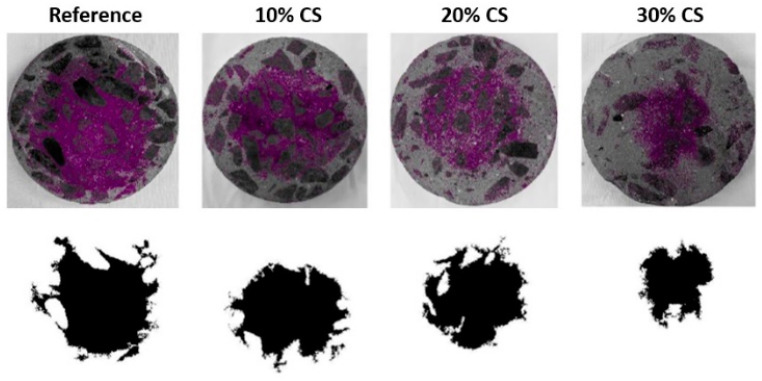
Determination of carbonation of concrete–CS with phenolphthalein solution at 28 days of exposure (1.2% CO_2_, 23 °C, and 65% RH).

**Figure 7 materials-16-06677-f007:**
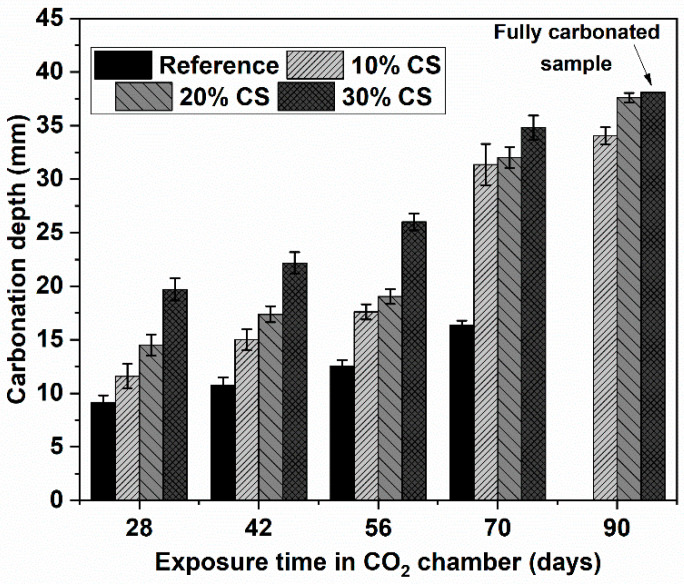
Carbonation depth of CS concrete.

**Figure 8 materials-16-06677-f008:**
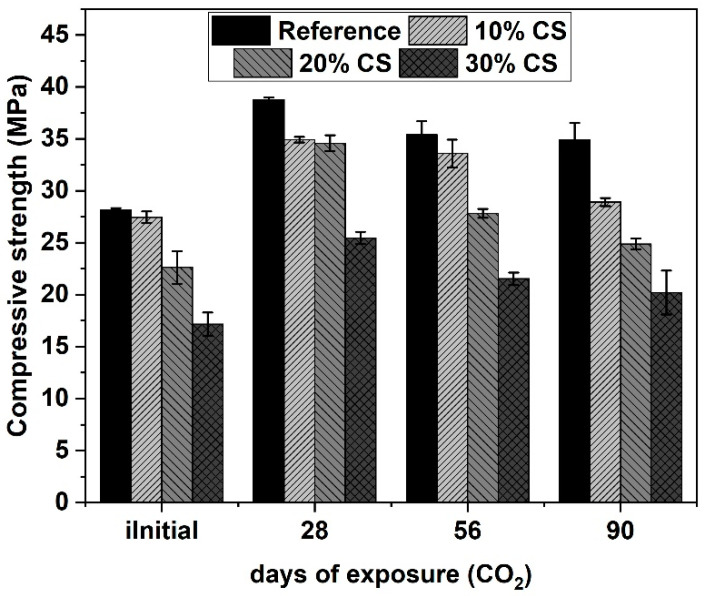
Compressive strength of concretes exposed to accelerated carbonation (1.2% CO_2_, 65% RH, 23 °C).

**Figure 9 materials-16-06677-f009:**
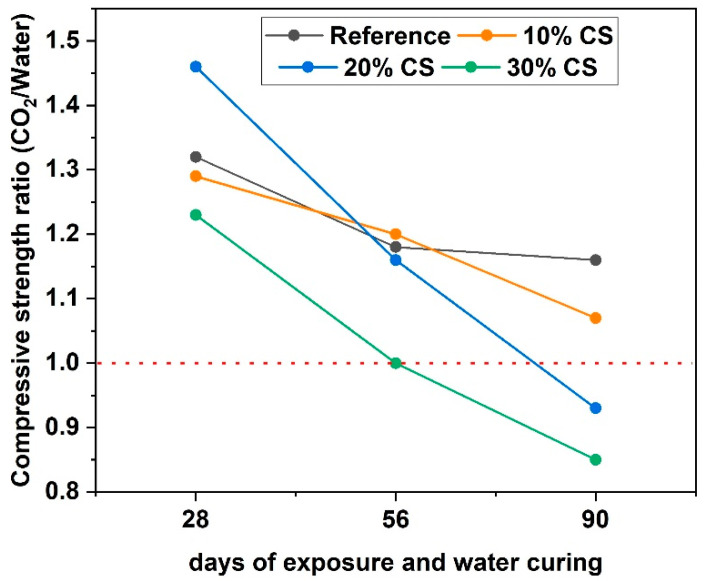
Compressive strength ratio of samples exposed to CO_2_ and cured in water.

**Figure 10 materials-16-06677-f010:**
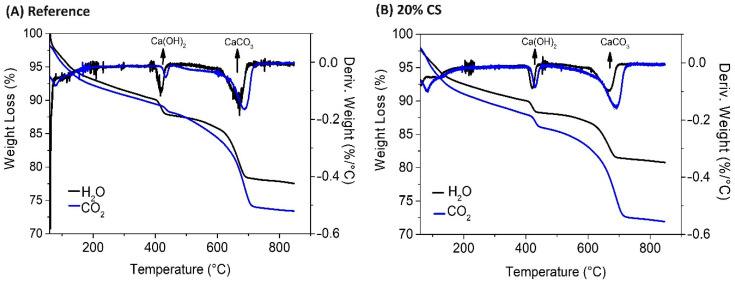
TG (**A**) and DTG (**B**) curves of the non-carbonated and carbonated reference and 20% CS mixtures.

**Figure 11 materials-16-06677-f011:**
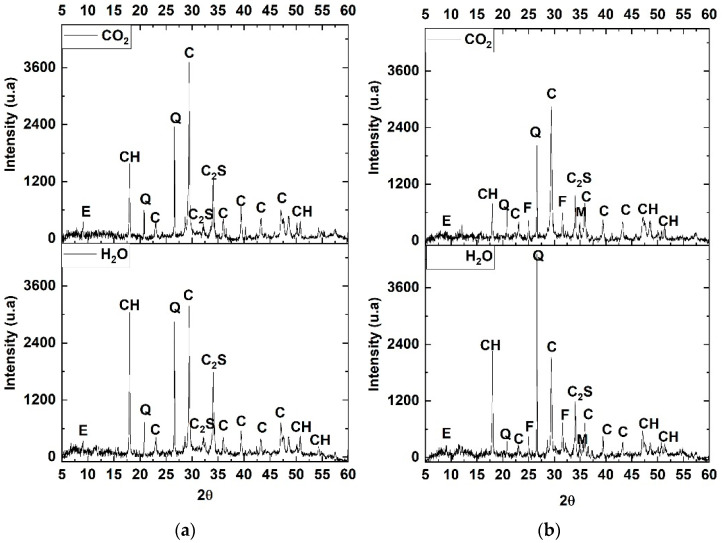
XRD patterns of (**a**) reference mixture and (**b**) 20% CS mixture. E: ettringite, CH: portlandite, Q: quartz, C: calcite; C_2_S: belite; F: fayalite, and M: forsterite.

**Figure 12 materials-16-06677-f012:**
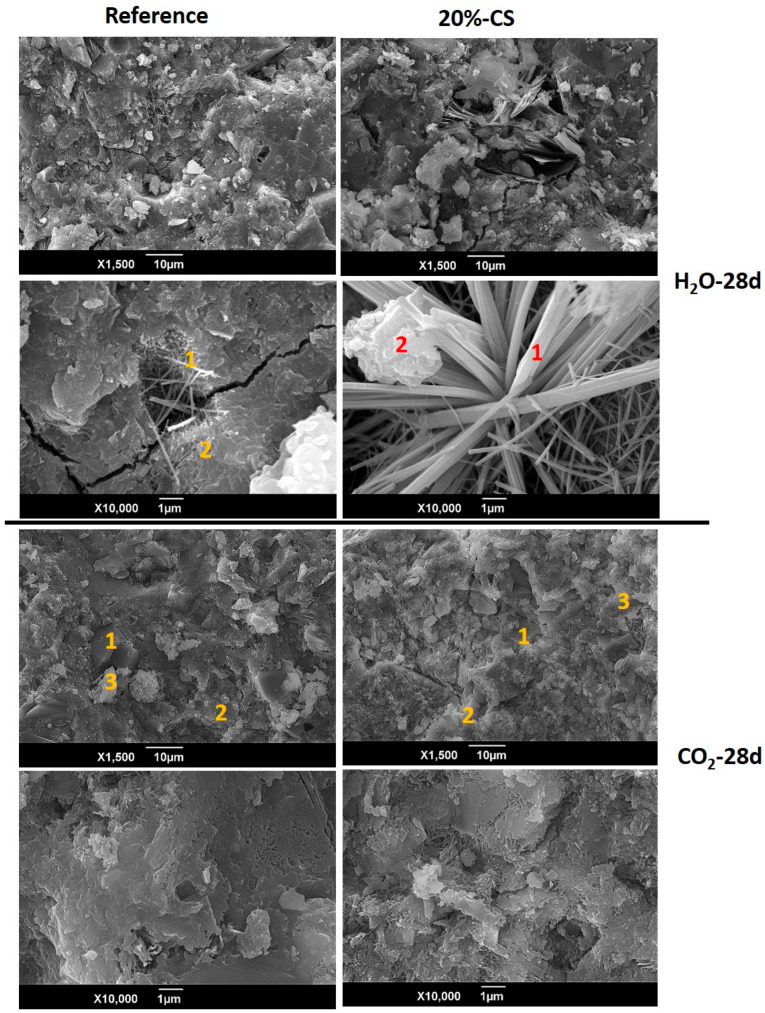
SEM and EDS points for reference and 20% CS pastes.

**Table 1 materials-16-06677-t001:** Chemical composition of cement and CS (% of oxides by mass).

Materials	SiO_2_	Al_2_O_3_	CaO	Fe_2_O_3_	SO_3_	Na_2_O	MgO	Cl	Other	LOI *
Cement	20.43	3.83	53.83	4.92	3.18	0.99	0.92	0.46	1.05	10.4
CS	28.88	6.14	2.70	55.58	0.71	1.00	0.86	-	4.11	0.02

* LOI: Loss on ignition.

**Table 2 materials-16-06677-t002:** Physical characterization of aggregates.

Materials	Apparent Density (kg/m^3^)	Fineness Modulus	Water Absorption (%)
Sand	2520	1.95	1.64
Gravel	2780	6.23	1.92

**Table 3 materials-16-06677-t003:** Mixture proportion of concrete with CS.

Mixture	Reference	10% CS	20% CS	30% CS
Cement (kg/m^3^)	360	324	288	252
CS (kg/m^3^)	-	41.27	82.54	123.81
Water (kg/m^3^)	180	180	180	180
Sand (kg/m^3^)	874	874	874	874
Gravel (kg/m^3^)	964	964	964	964
SP (kg/m^3^)	1.2	1.2	1.2	1.2

**Table 4 materials-16-06677-t004:** Water absorption, apparent density, and voids of concretes.

Mixture	Water Absorption (%)	Apparent Density (kg/m^3^)	Permeable Voids (%)
Reference	9.12	2676	19.59
10% CS	8.39	2756	18.85
20% CS	9.51	2785	20.97
30% CS	10.76	2852	23.70

**Table 5 materials-16-06677-t005:** % Weight loss in non-carbonated and carbonated mixture.

Sample	Weight Loss (60–450 °C), %	Total Weight Loss (60–850 °C), %
Reference–H_2_O	12.23	22.46
Reference–CO_2_	11.79	26.59
20% CS–H_2_O	11.76	19.21
20% CS–CO_2_	13.96	28.07

**Table 6 materials-16-06677-t006:** EDS results for reference and 20% CS pastes.

		Element (Weight %)								
Medium		Point	C	O	Al	Si	S	Ca	Fe	Ca/Si Molar Ratio
H_2_O	Reference	1	9.08	54.14	0.96	6.95		28.86		2.91
		2	7.84	49.90		7.34		34.92		3.33
	20% CS	1	7.98	62.21	3.30	3.54	1.71	21.25		4.21
		2	9.51	64.21	1.53	5.68		19.08		2.35
CO_2_	Reference	1	11.41	58.59		1.98		28.02		9.92
		2	9.88	57.79	1.16	6.72		24.45		2.55
		3		41.21	2.93	7.02	3.40	45.45		4.54
	20% CS	1	6.67	54.75		37.10		1.49		0.03
		2	11.20	54.17	1.41	9.69		21.34	2.19	1.54
		3	12.11	49.90	0.71	3.99		30.54	2.74	5.36

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
