# Peer review of "Exploring the Potential of Alternative Materials in Concrete Mixtures: Effect of Copper Slag on Mechanical Properties and Carbonation Resistance"

_materials, 2023, doi:10.3390/ma16206677_

Round 1

Reviewer 1 Report

The manuscript presents an interesting work about the production of concrete mixtures using copper slag (CS) waste to substitute part of cement for sustainability purposes. The manuscript reports the methodology for fabricating concrete mixtures with CS, characterization methods and results. This is a good piece of work that is well-written and with novel results suitable for Materials. The results are generally clear and well-discussed; however, the manuscript needs improvement. The following comments should be addressed before the manuscript can be recommended for publication:

Introduction

-While the authors present a literature review about the use of CS to fabricate asphalt and concrete, the literature review lacks a description of the results of the papers that deal with the use of CS in concrete. They mentioned that some studies are contradictory; however, details about this contradiction are not provided. The authors should provide 3-5 relevant references of CS in concrete, describing and discussing the results from these references, which is important for this work.

-In the last paragraph (Section 1.2), the authors should emphasize what is new and unique in this work that has not been done before based on the presented literature, i.e., what is the novelty? How is this work different?

Section 2

-Please provide the manufacturer details of the cement used, i.e., technical data sheet or website from the manufacturer.

-CT and FA in the title of Table 1 have not been defined in the text.

-Figure 2 is blurred. Please provide a better image.

Section 3

-At the end of Section 3, the authors should add a subsection with a discussion based on the many results presented and further discuss the applications and sustainability. They could also compare their materials and results with other results from the literature. Also, the authors should highlight the advantages and disadvantages of using CS on concrete. In addition, some estimation about the cost savings by using CS instead of cement could be added.

Minor editing of English language required.

Author Response

The authors are grateful for the reviewer's comments and suggestions. We addressed each individual point and we made modifications to the text to improve the general flow of the manuscript. To facilitate the review process, we marked in blue all the new and modified text that addresses the reviewers’ points.

Reviewer #1:

The manuscript presents an interesting work about the production of concrete mixtures using copper slag (CS) waste to substitute part of cement for sustainability purposes. The manuscript reports the methodology for fabricating concrete mixtures with CS, characterization methods and results. This is a good piece of work that is well-written and with novel results suitable for Materials. The results are generally clear and well-discussed; however, the manuscript needs improvement. The following comments should be addressed before the manuscript can be recommended for publication:

Introduction

-While the authors present a literature review about the use of CS to fabricate asphalt and concrete, the literature review lacks a description of the results of the papers that deal with the use of CS in concrete. They mentioned that some studies are contradictory; however, details about this contradiction are not provided. The authors should provide 3-5 relevant references of CS in concrete, describing and discussing the results from these references, which is important for this work.

Author´s response:

Thank you very much for the suggestion. Contradictory studies are mentioned a bit more, and additional references have been included to expand the introduction on the use of CS.

Choudhary[22] used replacement volumes between 20% and 100% of the natural coarse aggregate with CS, observing a decrease in compressive and tensile strength of 39% and 37.5%, respectively. However, when Lori [23] used CS as coarse aggregate, compressive strength increased by 22% when 100% of the natural coarse aggregate was substituted.

Research on CS as a supplementary cementitious material (SCM) has garnered attention recently due to the importance of decarbonizing the cement industry [24]. For example, Vizcaino & Silva [25] used CS as an SCM in a preliminary study on mortars, observing a decrease in compressive strength at all percentages used (10%-50% by volume) at early ages (7 and 28 days of curing). However, at 150 days of curing, the mortar with 10% CS showed an 8.1% increase compared to the reference mixture. He et al., [26] reported the application of CS as a partial replacement (10–50 wt%) for Portland cement in pastes. They also noted a loss of compressive strength of 85% and 33% at 3 and 28 days, respectively, in the paste with 50% CS compared to the composite without CS.

References

[22] S. Choudhary, P. Ravi Kishore, and S. Pachaiappan, “Sustainable utilization of waste slag aggregates as replacement of coarse aggregates in concrete,” Mater Today Proc, vol. 59, pp. 240–247, 2022, doi: 10.1016/j.matpr.2021.11.103.

[23]       A. R. Lori, A. Hassani, and R. Sedghi, “Investigating the mechanical and hydraulic characteristics of pervious concrete containing copper slag as coarse aggregate,” Constr Build Mater, vol. 197, pp. 130–142, 2019, doi: 10.1016/j.conbuildmat.2018.11.230.

[24]       R. Snellings, P. Suraneni, and J. Skibsted, “Future and emerging supplementary cementitious materials,” Cem Concr Res, vol. 171, no. April, 2023, doi: 10.1016/j.cemconres.2023.107199.

[25]       G. Vizcaino Méndez, Y.F. Silva-Urrego, “cobre como material cementicio suplementario en morteros Influence of copper slag as supplementary cementitious material in mortar,” pp. 1–6, 2023.

[26]       R. He, S. Zhang, X. Zhang, Z. Zhang, Y. Zhao, and H. Ding, “Copper slag: The leaching behavior of heavy metals and its applicability as a supplementary cementitious material,” J Environ Chem Eng, vol. 9, no. 2, 2021, doi: 10.1016/j.jece.2021.105132.

-In the last paragraph (Section 1.2), the authors should emphasize what is new and unique in this work that has not been done before based on the presented literature, i.e., what is the novelty? How is this work different?

Author´s response:

Thank you very much for the suggestion. The last paragraph of section 1. 2 has been modified.

This study presents the influence on the properties in both fresh and hardened states of concretes with different levels of replacement (0%-30% by volume) of CS by Portland cement in concrete. The properties in fresh (slump) and hardened state (compressive strength, density, absorption, porosity, carbonation resistance) were evaluated. Simultaneously, a microstructural analysis was conducted using thermogravimetric (TGA-DTG), Scanning Electronic Microscopy (SEM), and X-ray diffraction patterns (XRD). The results of this study will contribute to the generation of knowledge obout the use of CS in concrete and its performance in environments with high concentrations of CO2, which are becoming increasingly common in our environment.

Section 2

-Please provide the manufacturer details of the cement used, i.e., technical data sheet or website from the manufacturer.

Author´s response:

Thank you very much for the suggestion. Section 2 the name of the cement manufacturer was included.

Hydraulic cement of general use of the company Argos

-CT and FA in the title of Table 1 have not been defined in the text.

Author´s response:

Thank you very much for your comment

The title of Table 1 was modified

Table 1. Chemical composition of Cement and CS (% of oxides by mass)

-Figure 2 is blurred. Please provide a better image.

Author´s response:

The figure has been improved.

Section 3

-At the end of Section 3, the authors should add a subsection with a discussion based on the many results presented and further discuss the applications and sustainability. They could also compare their materials and results with other results from the literature. Also, the authors should highlight the advantages and disadvantages of using CS on concrete. In addition, some estimation about the cost savings by using CS instead of cement could be added.

Author´s response:

We greatly appreciate your valuable comments and suggestions

Regarding the request to add a discussion subsection at the end of section 3, we appreciate your interest in a more detailed discussion of the results and sustainable applications of our work. For this reason, some comparisons with other studies are added in this section. Although it is important to note that there are few studies on the use of CS as a supplementary cementitious material.

However, in Section 3.4, the following was added:

This behavior was similar to that observed by Silva and Delvasto [40] when they used residue of masonry (RM) as SCM in self-compacting concrete and subjected it to accelerated carbonation. It was evident that with higher percentages of cement replacement, carbonation increased, attributed to the lower alkaline reserve of the concrete, as well as the low reactivity of the RM.

In Conclusion

The incorporation of CS in low percentages as a SCM (around 10%) improves flowability, enabling the production of concrete mixtures with a lower water-cement ratio while achieving the same consistency as a concrete mixture with 100% cement. This could compensate for the slight reduction in compressive strength at early ages, which is not very significant at longer curing ages.

Reviewer 2 Report

In this paper, concrete with different copper slag replacement ratios was investigated on their flowability, compressive strength, absorption, and carbonation resistance. The microstructure was analyzed by X-Ray diffraction, scanning electron microscope, and thermogravimetric analysis. Several useful conclusions were drawn. The use of copper slag was an efficient method to reduce CO2 emissions for the reducing of cement usage. The article was valuable for waste recycle and environmental protection. The authors’ efforts should be encouraged. My comments were list as following.

(1) Abstract. The workability, compressive strength, carbonation resistance were decreased as the CS content increased. How did the authors get the conclusion “CS can be used as a promising substainable alternative in concrete and the management of this residue.” Please clarify it.

(2) The optimal replacement ratio of CS should be recommended for practical usage according to the experimental results.

(3) Line 92. What was the meaning of GU?

(4) Line 107. The abbreviation of LOI refers to?

(5) Figure 2. Particle size distribution of cement was missed, please supplement.

(6) Lines 235-238. The sentence make reviewer confused please rewrite it.

(7) Line 296. The subtitle of 3.5 was missed.

(8) Line 341. What did EC indicate?

Author Response

Response to reviewer

The authors are grateful to the reviewers’ comments. We addressed each individual point and we made modifications to the text to improve the general flow of the manuscript. To facilitate the review process, we marked in blue all the new and modified text that addresses the reviewers’ points.

I attach the word file responding to each of the comments and suggestions.

Regards

Round 2

Reviewer 1 Report

The authors have addressed the reviewer's comments. I recommend the paper for publication.